# Acetylcholine and Its Receptors in Honeybees: Involvement in Development and Impairments by Neonicotinoids

**DOI:** 10.3390/insects10120420

**Published:** 2019-11-22

**Authors:** Bernd Grünewald, Paul Siefert

**Affiliations:** Institut für Bienenkunde, Polytechnische Gesellschaft, FB Biowissenschaften, Goethe-Universität Frankfurt am Main, Karl-von-Frisch-Weg 2, D-61440 Oberursel, Germany; siefert@bio.uni-frankfurt.de

**Keywords:** insecticides, *Apis mellifera*, brood, larvae, nicotinic acetylcholine receptors, royal jelly

## Abstract

Acetylcholine (ACh) is the major excitatory neurotransmitter in the insect central nervous system (CNS). However, besides the neuronal expression of ACh receptors (AChR), the existence of non-neuronal AChR in honeybees is plausible. The cholinergic system is a popular target of insecticides because the pharmacology of insect nicotinic acetylcholine receptors (nAChRs) differs substantially from their vertebrate counterparts. Neonicotinoids are agonists of the nAChR and are largely used in crop protection. In contrast to their relatively high safety for humans and livestock, neonicotinoids pose a threat to pollinating insects such as bees. In addition to its effects on behavior, it becomes increasingly evident that neonicotinoids affect developmental processes in bees that appear to be independent of neuronal AChRs. Brood food (royal jelly, worker jelly, or drone jelly) produced in the hypopharyngeal glands of nurse bees contains millimolar concentrations of ACh, which is required for proper larval development. Neonicotinoids reduce the secreted ACh-content in brood food, reduce hypopharyngeal gland size, and lead to developmental impairments within the colony. We assume that potential hazards of neonicotinoids on pollinating bees occur neuronally causing behavioral impairments on adult individuals, and non-neuronally causing developmental disturbances as well as destroying gland functioning.

## 1. Introduction

Acetylcholine (ACh) is an evolutionary highly-conserved signaling molecule. It preceded the appearance of the nervous system since it is expressed in bacteria, archaea, in eucaryotic unicellular organisms, and in higher organisms such as plants, fungi, and animals [1]. Therefore, the neuronal system basically utilizes the existing cholinergic system and improved the communication speed by releasing ACh from vesicles during synaptic transmission. However, the non-neuronal cholinergic system remains side-by-side to the neuronal cholinergic system within animals. The functional principles of both systems are basically similar. They comprise of choline acetyltransferase (ChAT) to synthesize ACh, receptors for ACh of the muscarinic (mAChR) and the nicotinic type (nAChR), ACh-degrading esterases (AChE) and choline transporters (ChT) for the uptake of choline after ACh degradation. In addition to its expression in the nervous system, these components have been widely localized in epithelial and endothelial tissues [2], in reproductive organs [3], and in muscle and immune cells. Thus, numerous cell functions can be regulated by ACh, such as gene expression, proliferation, differentiation, cytoskeletal organization, cell–cell contact, locomotion, migration, ciliary activity, electrical activity, secretion, and absorption [4].

Compared to vertebrates, the non-neuronal cholinergic system of insects is largely understudied, although it is crucial during all developmental stages, and ACh, AChE, and ChAT are present in very much higher titers than in nervous tissues [5]. In most insect species, two AChE are present, and a large group of insecticides specifically target those esterases, such as organophosphates and carbamates [6]. In *Apis mellifera* one of the AChEs is membrane-bound and found in the CNS (AmAChE2), while the other is soluble (AmAChE1) and additionally found in the thorax, abdomen, and leg in non-neuronal tissue and the peripheral nervous system [7]. Apparently, the amount of the soluble AmAChE1 is regulated by the breeding activity of honey bee colonies, which provides further evidence for the influence of the cholinergic system on reproduction in insects [8]. In *Tribolium castanaeum,* expression of the AChE gene TcAce2 is important during female reproduction, embryo development, and offspring growth [9]. In *Drosophila melanogaster*, which has only one AChE gene, a non-neuronal effect was reported in which ACh, after its transport through the hemolymph, regulates the heart rate [10]. The importance of ACh for the insect immune responses is suggested since cholinergic disrupting chemicals impair immune responses [11,12]. Furthermore, initial evidence shows that hemocytes and the fatbody express nAChRs subunits in bees [13].

Cholinergic synaptic transmission has been intensively studied as it is prevalent within the insect brain. Nicotinic acetylcholine receptors have been localized in most brain neuropils, and functional nAChRs were characterized in vitro in various species (cf. [14] for review). Insect nAChRs are pentameric ionotropic receptors and cation channels with a high Ca^2+^ permeability (e.g., [15,16]). Their physiologies are in accordance with a function during excitatory synaptic transmission within the insect brain. Their molecular and functional similarities (e.g., high Ca^2+^-permeability and sequence homologies) to the vertebrate neuronal nAChRs imply that they also mediate modulatory functions. Accordingly, cholinergic signal transduction is required for olfactory learning and memory formation in insects (reviews: [17,18,19]). Honeybee nicotinic receptors share many features of insect nAChRs with respect to localization [20], pharmacology [21,22,23], permeability [15], and molecular identity [24,25,26]. The pharmacology differs substantially from its vertebrate counterparts [27,28]. Therefore, neonicotinoids have been developed that specifically target insect nAChR-dependent synaptic transmission, acting agonistically on the receptor with high specificity [29,30] (see also: [31]).

Neonicotinoids are a widely used group of insecticides. Not surprisingly, one of the problems with agricultural neonicotinoid applications is that they also bind to cholinergic receptors and induce currents through neuronal nAChRs [23,32,33,34] of pollinating insects, such as honeybees or bumblebees, and impair cholinergic transmission and—as a consequence—behavioral output. Socially living bees may be affected in several ways. Firstly, as adults, during foraging, during trophallactic contact, or by consumption of stored contaminated nectar, honey, or pollen. Secondly, as larvae after been fed with contaminated brood food. Therefore, bees need to be protected from exposure to neonicotinoids. Many studies investigated survival after insecticide treatment targeting the cholinergic system [35,36,37,38,39,40,41,42], and numerous studies have examined their effect on brood development by registering colony constitutions of honeybees (e.g., [43,44,45]) or bumblebees (e.g., [46,47,48]). Furthermore, sub-lethal neonicotinoid effects on adult honeybees comprise disturbances of navigation and orientation [49,50,51], walking behavior [52], learning and memory [53,54], foraging behavior [55], and nurse-larva-interactions [56]. For comprehensive overviews on sub-lethal neonicotinoid effects on honeybees, see [57,58,59,60]. Despite the fact that field studies largely failed to unambiguously demonstrate adverse effects of treated crop fields on whole hives the behavioral experiments clearly indicate impairments on individually treated honeybees. It is plausible to assume that the reported deficits also occur in those hives that appear vital upon superficial inspection.

The mode of action through which neonicotinoids induce these effects may be manifold. Certainly, acting via neuronal nAChRs is one of the major routes. In addition to this mechanism, neonicotinoids may affect muscarinic AChRs that are, as yet, largely uncharacterized in bees. A third way, which we assume to be particularly important for developmental effects, is via non-neuronal AChRs. Here, we review the properties of honeybee nAChRs and the actions of neonicotinoids on the neuronal receptors. We will then discuss the effects of neonicotinoids on larval and adult development and present an integrative model of cholinergic signaling and disturbances by neonicotinoids.

## 2. Acetylcholine Receptors in the Honeybee

Acetylcholine is the major excitatory transmitter in the insect brain (reviews in [14]). Immunolabelling of the nAChR or ChAT, as well as the histochemistry of AChE activity and in situ hybridization studies of the various nAChR α-subunits identified several pathways and neuropils that are presumably cholinergic in insects. In bees, the olfactory system and the visual neuropils probably rely mainly on cholinergic signal transmission [20]. Axons of the olfactory receptor neurons probably release ACh onto postsynaptic neurons within the antennal lobes (ALs), and a subpopulation of projection neurons from the AL form cholinergic synapses with Kenyon cells within the mushroom body (MB) lip regions. Honeybee AL neurons, as well as Kenyon cells, stain against nAChR antibodies [20]. The lamina, medulla, and lobula of bees contain cholinergic neurons as well as neurons of the central complex.

The honeybee nAChR is an ionotropic receptor of the cys-loop receptor family, a pentameric receptor whose stoichiometry is as yet unknown [26]. Sequence analyses identified nine different α-subunits, Amelα1–9, and two β-subunits, Amelβ1–2 [24,25,61]. Amelα5, Amelα7, and Amelα8 are expressed in MB Kenyon cells and in AL neurons. Amelβ2 subunits are found in Kenyon cells. In the optic lobes, Amelα2, Amelα3, and Amelα7–2, expressions were identified [24,25]. The native honeybee nAChRs in Kenyon cells and AL neurons are cation-selective channels with a neuronal pharmacological profile. Pressure applications of ACh or nicotine induce rapidly activating inward currents in cultured bee neurons [15]. The nAChR of Kenyon cells has a high Ca^2+^-permeability [15,62], and calcium imaging experiments in vitro revealed a strong intracellular Ca^2+^ signal during the application of nicotinic agonists [63,64]. The nAChR, therefore, mediates membrane depolarization and the direct influx of Ca^2+^ into the postsynaptic neuron upon activation. Currents through nAChR are blocked by nicotinergic blockers curare, methyllycaconitine, dihydroxy-β-erythroidine, hexamethonium, and mecamylamine [15,21,22,23]. ACh, as well as carbamylcholine, are full agonists, whereas nicotine, epibatidine, and cytisine are partial agonists [21]. Despite its neuronal profile, the honeybee nAChR has a rather unusual pharmacology as compared to its vertebrate counterparts. Since atropine blocks ACh-induced currents, a “mixed” pharmacology for the insect nAChR was suggested by some authors [28,65]. However, muscarinic agonists muscarine, pilocarpine, or oxotremorine do not induce currents through honeybee nAChRs [21]. Finally, the GABA (γ-aminobutyric acid) receptor blockers picrotoxin, bicuculline, and fipronil, as well as the glycine receptor blocker strychnine, act antagonistically [21,22]. Neonicotinoids are agonists of the insect nAChRs (review: [66]). Due to their low human toxicity and their relative specificity to insect over vertebrate nAChR, they represent a commercially very successful insecticide group [67,68]. Imidacloprid is a partial agonist of the honeybee nAChR (Kenyon cells: [23,33,34]; antennal lobe neurons: [22,32,62]), and clothianidin acts as a full agonist [34].

Behavioral pharmacological studies indicate that nAChR are involved during various phases of classical conditioning, memory formation, and retrieval (review: [18,19]). However, the effects caused by nicotinic antagonists are complex and often contradictory. Injections of the nAChR antagonists mecamylamine, α-bungarotoxin (BGT), or methyllylcaconitine (MLA) into the honeybee brain impaired acquisition (mecamylamine, [69]) or long-term memory (BGT, MLA, [70]). Interestingly, odor information processing appears to be largely unaffected by pharmacological treatments since odor learning, per se, is not impaired while imidacloprid perfusions diminish odor signals in antennal lobe glomeruli [71]. That may indicate that the native nAChR within the bee brain in vivo differs from the nAChR investigated in vitro. At least several different receptors with differing pharmacologies (probably also different stoichiometries) are expressed in honeybees. Given that various blockers target different nAChR subtypes, it was assumed that at least two nAChR (one BGT-sensitive and one BGT-insensitive nAChR) are differentially involved during olfactory learning and memory formation in bees (e.g., [70]; review: [18]). Therefore, care is needed by assigning the effects of various drugs or insecticides to certain nAChRs. Unfortunately, pharmacological experiments on the functions of nAChRs within the honeybee visual system are missing, although the optic lobes express nAChRs.

Muscarinic acetylcholine receptors are G protein-coupled receptors with seven transmembrane domains. In humans, five mAChRs (m1–m5) have been characterized that are expressed not only in the peripheral and central nervous system but also in epithelial (airway, skin, intestine, ovary, urothelium), endothelial (pulmonary vessels), immune, and mesenchymal (fibroblasts, tenocytes, smooth muscle fibres) cells [2]. The insect mAChRs are less well-studied. Two mAChRs were cloned and described in *D. melanogaster*, and *T. castaneum* (an A- and B-type), both activated by ACh but have different sensitivities to muscarine and binding of atropine and scopolamine. Both receptors have been identified in all arthropods with a sequenced genome [72]. Recently, a third (C-type) mAChR family has been described in *D. melanogaster* [73].

## 3. Acetylcholine in Bee Development

In several vertebrate tissues, ACh demonstrates a proliferative, trophic effect via nicotinergic and muscarinic receptors [3]. In insects, the cholinergic system is crucial during all developmental stages, and ACh, AChE, and ChAT are present in very much higher titers as compared to vertebrates [5]. Apart from its occurrence in honey and bee bread [74,75,76,77,78], bees apparently feed ACh to developing larvae, as it was found in millimolar concentrations in larval food [76]. This recent study confirms and extends earlier studies reporting surprisingly high ACh concentrations in brood food (see below).

Worker larval nutrition is categorized as “worker jelly” and “modified worker jelly”, indicating a general shift in protein, sugar, and lipid contents [79] around day three of larval development [80]. This shift has also been reported for ACh content in worker nutrition. While larvae below 5 mg weight receive a relatively high ACh amount in their food (1.1 mg free base per gram dry larval food [81]; erratum: ‘Die Naturwissenschaften 47, p. 456, 1960’) food for larvae weighing between 10 and 20 mg contains less ACh (0.73 mg g^−1^). The oldest larvae received modified worker jelly with the least amount of ACh (0.16 mg g^−1^) [81]. This is generally consistent with the study by Wessler et al. (2016) reporting 4.13 mM (estimated 0.72 mg g^−1^) ACh in worker jelly if the developmental state of ‘larger larvae with visible food’ [76] corresponds to 10 to 20 mg weighting larvae. Drone food also contains relatively high ACh concentrations (1.8, 1.65, and 0.66 mg g^−1^ for drone larvae weighing <5, 10–30, and >30 mg, respectively) [81].

The reduction in the ACh content during worker development is conceivable since gland secretion decreases in favor of sugar containing food from the honey stomach [79], and ACh is synthesized in hypopharyngeal canal cells via membrane-bound ChAT [76]. The synthesis during jelly excretion and the surrounding acidity of pH 4.0 makes ACh very stable in larval honey bee food [82] because AChE is not enzymatically active under such acidic conditions. ACh in brood food can even be preserved after two hours of boiling in water [74].

Royal jelly also contains high ACh amounts. It is fed to developing honey bee queens and, compared to worker nutrition, contains a higher amount of sugar [79]. According to [81], ACh content decreases from 1.7 to 1.1 mg g^−1^ in royal jelly in cells of young (weight <5 mg) and old (>25 mg) larvae, respectively. This represents a 35% decrease during queen development compared to an 85% decrease during worker development and may influence caste determination. HPLC analyses quantified 8 mM (1.4 mg g^−1^) ACh in freshly isolated royal jelly (2–3 h after the nursing of fertilized eggs) and 4.64 mM (estimated 0.81 mg g^−1^) in commercially available royal jelly [76]. Experimentally reducing the ACh content in artificial brood food increased larval mortality [76]. ACh-uptake by larvae is, therefore, required for the proper development of queens, workers, and drones. It is produced from non-neuronal tissue and probably acts via non-neuronal AChRs.

## 4. Neonicotinoids Affect Larval and Adult Development

Although ACh is important during larval development, only a few studies exist that have investigated neonicotinoid effects on the bee ontogeny. However, most of the available studies investigating larval development describe a developmental delay and some abnormal appearances. Also, there is reason to assume that some effects occur in those adults that were exposed to neonicotinoids as larvae (e.g., [83,84]). Repeated administration of 0.2–20 mg L^−1^ thiamethoxam in artificial *Apis mellifera* rearing experiments caused more brownish larvae, delayed pupation time, and some larvae failed in eclosion during metamorphoses [35]. Delayed development was also reported when the larvae were fed 5 µg kg^−1^ [85] or 10 mg L^−1^ imidacloprid. Furthermore, honeybee larvae reared in combs with insecticide residues, including 45 µg kg^−1^ imidacloprid, showed retarded development [86]. While 30 or 300 µg kg^−1^ imidacloprid fed to *Osmia lignaria* larvae delayed their development under field conditions, no effects were reported under laboratory conditions [87]. A recent study described a doubling of honeybee larval development time when fed 10 mg L^−1^ clothianidin and of total development time with 2 mg L^−1^ clothianidin in vitro. However, the pupal development time was not affected [40]. Video registrations of development within observation hives [56] showed that 200 µg kg^−1^ thiacloprid in sugar syrup fed to the colony prolonged the feeding timespan until capping of the cell by half a day. Similar effects were observed when colonies were fed 10 µg kg^−1^ clothianidin. Furthermore, high dosages of clothianidin (100 ppb) and thiacloprid (8.8 ppm) prolonged the development times of eggs and from larval hatch to cell capping within the colonies. Interestingly, clothianidin increased pupal development time, whereas thiacloprid decreased it [56]. In the stingless bee *Scaptotrigona aff. depilis* thiamethoxam decreased development time in vitro from 15 to 10 days (0.044 ng/larva) or 8 days (4.375 ng/larva) while pupal development increased from 12 to 18 and 17 days, respectively [39].

These in vitro insecticide experiments show that the cholinergic system is important for larval development and is disturbed by neonicotinoids. However, a direct delivery of neonicotinoids from nurse bees to larvae via brood food appears not to occur under field conditions. Virtually no pesticide residues were found in royal jelly, even when colonies were fed with high (75–800 g L^−1^ active ingredient) pesticide concentrations. Only 0.016% of the consumed thiacloprid reaches the secreted royal jelly [88]. This is consistent with distributions of other insecticides within workers. Low radioactivity was measured in hypopharyngeal glands after individuals were fed with radiolabelled carbaryl and diflubenzuron [89] or carbofuran and dimethoate [90]. If direct transmission by honey was the main reason for residues in royal jelly, worker larvae would receive very little neonicotinoids since royal jelly seems to contain more sugar than worker jelly [79]. However, this may increase as the older worker larvae are fed modified royal jelly. As several studies demonstrate delayed larval development within honeybee colonies after chronic neonicotinoid treatment (e.g., [56,76,86]), impairments of important nursing morphologies within workers may cause such effects rather than direct feeding of toxic insecticides to larvae. As a consequence, cholinergic transmission could also be important for the development and maturation of adult bees.

The cholinergic system continues to develop in the adult honeybee. During the first week after adult eclosion, the brain activity of AChE increases and remains at this level until old age [91]. Due to the plasticity of each individual worker, the bee colony as a whole is very adaptive in its division of labor and can shift nursing and foraging activity due to environmental and colony demands. This adaptation is under the influence of various internal (e.g., colony size, brood size, diseases) and external factors (e.g., nutrition, pollen supply, season or weather, stressors like pesticides) and involves nutritional stimuli communicated via food exchange (cf. [92] for review). The subsequent development or degradation of the hypopharyngeal glands [93] is hormone-dependent. Newly emerged workers have undeveloped glands with small acini, and with nursing activity, they increase in size and produce the protein-rich jelly. The glands decrease in size and activity again when the worker starts to forage. As the nursing workers feed the young, other nest mates and the queen [94,95], glandular ACh could be consumed by all individuals of the colony throughout their whole life span. Therefore, a regulatory influence of ACh on adult development and behavior is plausible. If the exchanged food contains differing amounts of ACh, cholinergic social signaling is likely to contribute to gland development and function and would explain the disturbing effects of neonicotinoids and other xenobiotics described below.

When newly emerged bees in cages were fed with 2 and 3 µg kg^−1^ imidacloprid in sugar and pollen pastry, respectively, the acini of hypopharyngeal glands were 14.5 and 16.3 percent smaller in diameter than the control after 9 days and 14 days of exposure, respectively [96]. Similarly, after one week of 1 ppb imidacloprid application in sugar syrup to colonies, 14-day old workers displayed reduced acini diameter sizes [97]. Similar effects were present when bees were exposed to 48 and 72 h treatment with 0.5 µg kg^−1^ imidacloprid in sugar solution [98] and in 10-day old bees after continuous feeding of 5 or 200 µg kg^−1^ imidacloprid under field and laboratory conditions [99]. Furthermore, caged honeybees chronically exposed to imidacloprid at LC_50_/5 in sugar and pollen showed reduced acini diameter sizes after 6, 9, and 14 days [100]. Comparable results were published for 8 and 12 days of exposure to 10 and 40 µg L^−1^ thiamethoxam [101]. In addition to these morphological abnormalities, neonicotinoid exposure also impairs hypopharyngeal gland function because acetylcholine secretion into the brood food diminishes after clothianidin or thiacloprid feeding [76]. This probably leads to brood impairments in small hives under semi-field conditions. Expression of Amelα3 and Amelα4 nAChR subunits in the hypopharyngeal glands indicates an ACh-dependency of the glandular function, although ChAT activity is not acutely blocked by neonicotinoids, but requires chronic exposure [76]. Obviously, other xenobiotics that interfere with cholinergic signaling in honeybees may lead to similar disruptions in glandular function. Among them are the acaricide coumaphos, an organophosphate that inhibits the AChE, and other insecticides like the carbamate fenoxycarb (for review see: [102]). Effects on hypopharyngeal glands were also reported for the GABA receptor blocker fipronil, the herbicide glyphosate, or the fungicide pyraclostrobin [102].

## 5. How Do Neonicotinoids Affect Honeybee Development?

Several mechanisms may underlie the delays in development, including impairments of the endocrine system, altered gene expression of metabolic pathways, and an increased energy use due to detoxification mechanisms. Some studies suggest that neonicotinoids interfere with the honey bee endocrine system [103,104,105]. Juvenile hormone slows down larval development, as ligaturing the corpora allata results in shortened worker ontogenesis [106]. Therefore, neonicotinoids may increase juvenile hormone titers in bees. However, so far, it has only been demonstrated in Lepidopterans that imidacloprid increases juvenile hormone titers in larvae and adult females of *Chilo suppressalis* [107] and that the corpora allata is under a cholinergic regulation in *Mythimna loreyi* [108]. Several gene expression studies report an upregulation of detoxification enzymes, such as cytochrome p450s, and effects on protein translation involved in metabolic pathways after insecticide treatment [99,109,110,111,112]. Honeybees show 10-fold or greater shortfalls in detoxification enzymes compared to *Drosophila melanogaster* or *Anopheles gambiae* [113], which may cause unspecific health deficits due to low detoxification capacities. Moreover, some studies report a general reduction in protein amounts in workers after neonicotinoid treatment [101,114,115] that promotes developmental delays. Imidacloprid and clothianidin were stated to alter protein, lipid, glucose, and glycogen levels and reduced bee body weight [105]. By contrast, newly emerged honeybees fed with nicotine up-regulate proteins involved in lipid, amino acid, glutathione, and nucleotide metabolism. The most up-regulated protein groups are related to energy metabolism and carbohydrate metabolism [116]. In larvae, proteins involved in energy and carbohydrate metabolism and developmental pathways were enriched after nicotine was fed in in vitro rearing experiments. Therefore, nicotine may promote larval growth [117]. These reports consistently show an influence of the cholinergic system on developmental processes within larvae and adult workers. The impairments of metabolic pathways could then result in morphological deficits in nurses and provoke reduced hypopharyngeal gland sizes that secrete less ACh into the queen, worker, and drone food.

## 6. Conclusions and Outlook

Neonicotinoid exposures affect honeybee vitality in various ways, behaviorally, morphologically, immunologically, and developmentally. These effects are mediated by neuronal and non-neuronal AChR. Figure 1 summarizes our current view concerning the developmental aspects of the cholinergic systems. We assume that ACh has a proliferative and/or trophic effect within honeybees and regulates gene expression, potentially via modulating juvenile hormone levels. Although evidence that juvenile hormone (JH) titers are affected by cholinergic pathways is weak, it is likely, because JH plays a central role in honeybee developmental and maturing processes. The ACh-induced gene expression affects metabolic pathways that control larval development and adult hypopharyngeal gland size and function. This, in turn, regulates ACh secretion from the glands with the described effects on larval development. Therefore, neonicotinoid effects can translate into disturbed colony development due to impairments of cholinergic systems in the offspring, nurses, or both.

We conclude that acetylcholine is a key signaling molecule in the individual bee, within the honeybee colony, and possibly in other social insect societies as well. As it acts on neuronal and presumably non-neuronal pathways and is socially transmitted between individuals, it is crucial for colony development and vitality. Interfering with this key molecule by insecticides causes multiple disturbances or unexpected side effects that may sum up to harmful threats even at low concentrations.

Obviously, several gaps exist in our knowledge of the mode of action of ACh signaling. While we know a lot about the *neuronal* AChR, hardly anything is published on the molecular identity or localization of the (various?) insect *non-neuronal* AChR. Such studies, however, are required to establish hypotheses on the interactions between ACh and the endocrine system, to understand the physiological mode of action of ACh during larval and adult development and, finally, to develop novel insecticides that are safer for beneficial insects.

## Figures and Tables

**Figure 1 insects-10-00420-f001:**
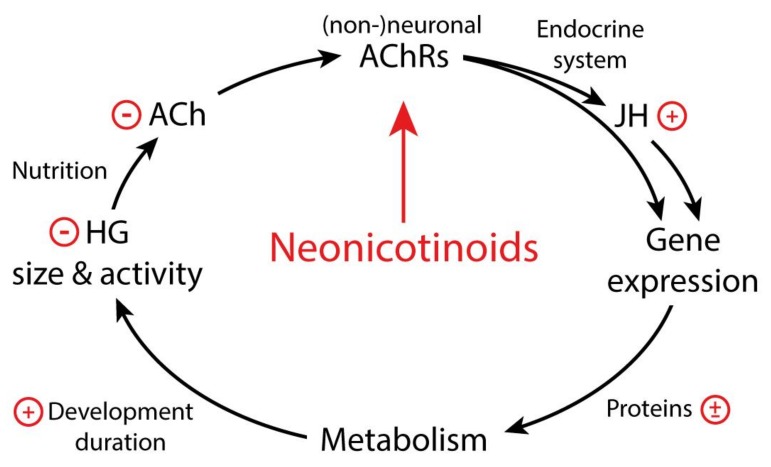
Metabolic and endocrine functions of acetylcholine (ACh) in honeybee adults and larvae and its disturbances by cholinergic pesticides, such as neonicotinoids. Impairments of neuronal or non-neuronal acetylcholine receptors (AChRs) of honeybees by neonicotinoids have been shown to increase (+) development duration of larvae and adults while reducing (−) hypopharyngeal gland (HG) size and its ACh secretion. This is likely to be a consequence of impaired energy and carbohydrate metabolism, preceded by up- or downregulation (±; via gene expression) of involved proteins, altering lipid, glucose, and glycogen metabolism. Therefore, ACh may directly or indirectly affect the endocrine system, as increased juvenile hormone (JH) titers result in developmental delays (see text for further details).

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
