# Peer review of "Acetylcholine and Its Receptors in Honeybees: Involvement in Development and Impairments by Neonicotinoids"

_insects, 2019, doi:10.3390/insects10120420_

Round 1

Reviewer 1 Report

This review paper describes neonicotinoid effects on both neuronal and non-neuronal AChRs of the honeybees. In addition, this paper explains possible mechanisms of adverse effects of neonicotinoid on larval development. It covers a lot of important literature that investigated neonicotinoid pharmacology. In my view, it is well organized and excellent to grasp the important information in this field. Also, English is well written and easy to understand. Therefore, I suggest that this review is acceptable for publication as this form. As far as reviews, I could not find any typos and wrong citation.

Author Response

Dear Reviewer,

thank you very much for you very positive review.

Sincerely

Bernd Grünewald

Reviewer 2 Report

The effects of neonicotinoid insecticides on honeybees have been interested in the public recently. The toxic effect of the neonicotinoids is based on their agonist binding to acetylcholine receptors of cholinergic systems in the insect. However, the cholinergic systems of honey bees did not well studied yet. Thus, this is a good topic to do a literature review.  The authors, Grünewald and Siefert, review the cholinergic systems of honey bees, including the acetylcholine receptors, signals, and its effects in bee development as well as the interaction with neonicotinoids pesticides. The findings are generally fine, and the topic should be interesting in readers and might help with future researches.

However, there are still some details that need to be clarified before publishing. Authors should explain and illustrate more details on the following matters:

The authors mentioned the “cholinergic systems” in the title. It is important to define the system and introduce functions of the system at the beginning (introduction part) for readers. Also, the title has a very board spectrum, but the contents are more focused on neonicotinoids, non-neuronal cholinergic system and how it effects on the development of bees. It will be better to modify the tile to meet the content.

Please describes details of the research progress of cholinergic systems in insects and even other animals. It already has good progress and working hypothesizes in the other insect and animal models. It is important to give a basic concept, information, and aspects of the system. It could help readers to understand what the research gaps at this system in honeybees.

Cholinergic systems include neuronal and non-neuronal systems. Both of them are important in health and responsible for different functions. The authors mentioned both neuronal and non-neuronal aspects in the manuscripts. It will be better to clearly indicate them separately.

Acetylcholine, acetylcholine-synthesizing enzymes, transporters, (muscarinic- and nicotinic-) receptors and degrading enzymes are all constructs in the non-neuronal cholinergic system. The authors mentioned some of them but not all. It is important to show the big picture and indicate the research progresses in honeybees in a review. If there is no honeybee research available, researches in other insect models should be mentioned.

Not only the neonicotinoids affect the cholinergic system. The organophosphate, carbamate, and pyrethroid pesticides are known to effects the cholinergic system. Authors should also be mentioned all possible pesticides rather than only mention neonicotinoids.

It is easier for readers to pick up the information summary from a figure. In section two, the authors mentioned many possible cholinergic projection and receptors in the bee brain. It will be better to add one or several schematic figures and a table to summary their locations and what signals and receptors involved. The table should also include refs.

The other sections should also include tables/figs summary.

In section 4, the authors mentioned, “most of the available studies describe a delay in larval development and some abnormal appearances”. It is not totally true. There are many studies indicated the larval stage pesticide exposure could lead to impartial learning in olfactory, aversive, and memory impair etc.

e.g.:

Yang, E. C., Chang, H. C., Wu, W. Y., & Chen, Y. W. (2012). Impaired olfactory associative behavior of honeybee workers due to contamination of imidacloprid in the larval stage. PloS one, 7(11), e49472.

Tan, K., Chen, W., Dong, S., Liu, X., Wang, Y., & Nieh, J. C. (2015). A neonicotinoid impairs olfactory learning in Asian honey bees (Apis cerana) exposed as larvae or as adults. Scientific Reports, 5, 10989.

The authors mentioned some abbreviations and terms without explanations. For example, Ln 59 mentioned the muscarinic the first time of AChRs. It is very easy to make readers confused science there no introduction to it before. Again, the authors might want to introduce the acetylcholine systems in the beginning. Ln94 mentioned the first time of the GABA receptor. Please at least provide a short explanation of its function and role. Also, for an abbreviation, authors should list the full name when the first time mentioned it. Authors should include an abbreviation table for all abbreviation mentioned in the manuscript.

Ln 203-214, authors mention the field and Lab conditions as well the alternations of hypopharyngeal gland size. It is also important to include that in the field condition, the natural food, ie phytochemicals, nutrition, and other environmental stresses (pesticides) might all affect the toxicity response and the size alterations in nurse hypopharyngeal glands of adult bees.

The scientific names of species should be italicized. Please correct at your references.

Please add a section or paragraph of future research direction.

Author Response

Dear Reviewer,

thanks a lot for your very valuable comments on our MS. Please find our responses to your points below. They helped to substantially increase the quality of our manuscript.

Yours Sincerely

Bernd Grünewald

---

Reviewer 2:

a) The authors mentioned the “cholinergic systems” in the title. It is important to define the system and introduce functions of the system at the beginning (introduction part) for readers. Also, the title has a very board spectrum, but the contents are more focused on neonicotinoids, non-neuronal cholinergic system and how it effects on the development of bees. It will be better to modify the tile to meet the content. b) Please describes details of the research progress of cholinergic systems in insects and even other animals. It already has good progress and working hypothesizes in the other insect and animal models. It is important to give a basic concept, information, and aspects of the system. It could help readers to understand what the research gaps at this system in honeybees. c) Cholinergic systems include neuronal and non-neuronal systems. Both of them are important in health and responsible for different functions. The authors mentioned both neuronal and non-neuronal aspects in the manuscripts. It will be better to clearly indicate them separately. d) Acetylcholine, acetylcholine-synthesizing enzymes, transporters, (muscarinic- and nicotinic-) receptors and degrading enzymes are all constructs in the non-neuronal cholinergic system. The authors mentioned some of them but not all. It is important to show the big picture and indicate the research progresses in honeybees in a review. If there is no honeybee research available, researches in other insect models should be mentioned.
e) Not only the neonicotinoids affect the cholinergic system. The organophosphate, carbamate, and pyrethroid pesticides are known to effects the cholinergic system. Authors should also be mentioned all possible pesticides rather than only mention neonicotinoids.

Thank you for these suggestions. We changed the title to better fit our content. Our MS is a rather conceptual review that brings to the reader’s attention a largely overseen role of ACh in insects. Due to this focus we wish to keep the review of the extensive literature on the animal cholinergic system to a minimum. The revised title better captures this intention. We, nevertheless, agreed to the reviewer suggestion to present a bigger picture and added a new section in the introduction to describe the current research status of the cholinergic system in model organisms. In this turn we clearly indicate the co-existence of the neuronal and non-neuronal system and list all of its components. Furthermore, we mention the organophosphates and carbamates to complete the picture. However, we are not aware of a paper describing specific pyrethroid effect on the cholinergic system. We’d be grateful if the reviewer could specify her or his comment here.

It is easier for readers to pick up the information summary from a figure. In section two, the authors mentioned many possible cholinergic projection and receptors in the bee brain. It will be better to add one or several schematic figures and a table to summary their locations and what signals and receptors involved. The table should also include refs.

The requested figure was published in Kreissl and Bicker 1989, which we cited. To avoid redundancy with an article by Cabirol and Haase just published as part of the Special Issue (Insects 2019, 10(10), 344), we wish to omit such an illustration in our review. Instead a citation of the mentioned article is inserted.

The other sections should also include tables/figs summary.

We respectfully disagree here with the reviewer. As mentioned above, we did not intend to provide a comprehensive review on the cholinergic system and neither on all aspects of neonicotinoid effects on bees. We rather refer the reader to the excellent reviews on these matters. Therefore, any additional figures would not illustrate the whole picture and would rather be misleading, because they cannot be complete in a very concise review.

In section 4, the authors mentioned, “most of the available studies describe a delay in larval development and some abnormal appearances”. It is not totally true. There are many studies indicated the larval stage pesticide exposure could lead to impartial learning in olfactory, aversive, and memory impair etc.

This is right. We rephrased this sentence to express more clearly that we mean the publications on neonicotinoid effects on development describe a developmental delay rather than something else (like shortening). In addition, we added the suggested papers to state a possible effect in adults after larval exposure.

The authors mentioned some abbreviations and terms without explanations. For example, Ln 59 mentioned the muscarinic the first time of AChRs. It is very easy to make readers confused science there no introduction to it before. Again, the authors might want to introduce the acetylcholine systems in the beginning. Ln94 mentioned the first time of the GABA receptor. Please at least provide a short explanation of its function and role. Also, for an abbreviation, authors should list the full name when the first time mentioned it. Authors should include an abbreviation table for all abbreviation mentioned in the manuscript.

We checked the whole text and corrected it accordingly.

a) Ln 203-214, authors mention the field and Lab conditions as well the alternations of hypopharyngeal gland size. It is also important to include that in the field condition, the natural food, ie phytochemicals, nutrition, and other environmental stresses (pesticides) might all affect the toxicity response and the size alterations in nurse hypopharyngeal glands of adult bees.
b) The scientific names of species should be italicized. Please correct at your references.
c) Please add a section or paragraph of future research direction.

We added these information and a few sentences on future research directions (last paragraph) accordingly and italicized scientific species names.

Round 2

Reviewer 2 Report

Reviewer: The authors have clearly introduced the acetylcholine system and the main purposes of this review. The introduction and title are also much clearer to the readers than the last version. However, there are still some minor suggestions which I would like to respond to and recommend the authors to modify.

Authors: Thank you for these suggestions. We changed the title to better fit our content. Our MS is a rather conceptual review that brings to the reader’s attention a largely overseen role of ACh in insects. Due to this focus we wish to keep the review of the extensive literature on the animal cholinergic system to a minimum. The revised title better captures this intention. We, nevertheless, agreed to the reviewer suggestion to present a bigger picture and added a new section in the introduction to describe the current research status of the cholinergic system in model organisms. In this turn we clearly indicate the co-existence of the neuronal and non-neuronal system and list all of its components. Furthermore, we mention the organophosphates and carbamates to complete the picture. However, we are not aware of a paper describing specific pyrethroid effect on the cholinergic system. We’d be grateful if the reviewer could specify her or his comment here.

These are two of the publications regarding the pyrethroid’s effect on the cholinergic system.

Burns, C. J., McIntosh, L. J., Mink, P. J., Jurek, A. M., & Li, A. A. (2013). Pesticide exposure and neurodevelopmental outcomes: review of the epidemiologic and animal studies. Journal of Toxicology and Environmental Health, Part B, 16(3-4), 127-283.

Eriksson, P., & Nordberg, A. (1990). Effects of two pyrethroids, bioallethrin and deltamethrin, on subpopulations of muscarinic and nicotinic receptors in the neonatal mouse brain. Toxicology and applied pharmacology, 102(3), 456-463.

The requested figure was published in Kreissl and Bicker 1989, which we cited. To avoid redundancy with an article by Cabirol and Haase just published as part of the Special Issue (Insects 2019, 10(10), 344), we wish to omit such an illustration in our review. Instead a citation of the mentioned article is inserted.

We respectfully disagree here with the reviewer. As mentioned above, we did not intend to provide a comprehensive review on the cholinergic system and neither on all aspects of neonicotinoid effects on bees. We rather refer the reader to the excellent reviews on these matters. Therefore, any additional figures would not illustrate the whole picture and would rather be misleading, because they cannot be complete in a very concise review.

This is right. We rephrased this sentence to express more clearly that we mean the publications on neonicotinoid effects on development describe a developmental delay rather than something else (like shortening). In addition, we added the suggested papers to state a possible effect in adults after larval exposure.

Reviewer: Ok.

The title of Section 4. “Insecticides on larval and adult development” seemingly covers many insecticides. However, the section mainly discusses the neonicotinoid, only a few other insecticides were included. In addition,  it seems to overlap with the topic of section 5. The authors might please reconsider and modify the titles of each section or add more section titles is fine. 

Ln 237-238  Authors mentioned ACh may play a role in this plasticity. Please provide more supported refs or explanations. Only the HPG size change and nurse bee might feed hive mates with jelly along with ACh can’t be valid pieces of evidence or supports.

Ln 251-3 “Besides these morphological abnormalities, neonicotinoid exposure also impairs hypopharyngeal gland function, because acetylcholine secretion into the brood food diminishes after clothianidin or thiacloprid feeding” Please provide refs.

Ln 241-242 “Several studies indicate that gland development can be regulated by the cholinergic system.” Please provide refs.

The division of labor has been correlated to several pheromones, including  JH, vitellogenin, and insulin. It will be good to take all of them into consideration and put them into your manuscript and model.

Ament, S. A., Wang, Y., & Robinson, G. E. (2010). Nutritional regulation of division of labor in honey bees: toward a systems biology perspective. Wiley Interdisciplinary Reviews: Systems Biology and Medicine, 2(5), 566-576.

There are other pesticides that can also induce the HPG alterations (review in Berenbaum et al 2019). Thus, they might affect the secretion of jelly and larvae development. Those pesticides and their effects would be good to mention in section 4.

 Berenbaum, M. R., & Liao, L. H. (2019). Honey Bees and Environmental Stress: Toxicologic Pathology of a Superorganism. Toxicologic pathology, 0192623319877154.

The Fig.1 model was too simplified and could confuse readers. Does the model represent the adults or larvae system? Where is AChE or AChT? Could the authors add more details that you have mentioned in the text? “Gene expression” is not clear enough. Please indicate which genes might be involved. Do they down-regulate or up-regulate? How they associate with physiological and behavioral consequences if any?

The evidence, so far, indicated that the neuronal and non-neuronal AChRs lead to different responses. Could the authors separate the models here?

 Second, “+” and “-” is not clearly indicated in the figure caption. Please add them.

The References and their format will need further proofreading. There are some errors. Here only a part of them is listed. Please double-check all references and their format

e.g., Ref. 2 and Ref. 73 were duplicated.

Ref. 2. Volume and issue missed.

Ref. 4 titles should use the sentence capital not capitalized all words.

Author Response

Responses to Reviewer 2:

These are two of the publications regarding the pyrethroid’s effect on the cholinergic system.
Burns, C. J., McIntosh, L. J., Mink, P. J., Jurek, A. M., & Li, A. A. (2013). Pesticide exposure and neurodevelopmental outcomes: review of the epidemiologic and animal studies. Journal of Toxicology and Environmental Health, Part B, 16(3-4), 127-283.
Eriksson, P., & Nordberg, A. (1990). Effects of two pyrethroids, bioallethrin and deltamethrin, on subpopulations of muscarinic and nicotinic receptors in the neonatal mouse brain. Toxicology and applied pharmacology, 102(3), 456-463.

Thanks a lot for the references. Nevertheless, we regard it not useful to include these articles in our MS, because (1) pyrethroids are not specific to AChR but rather are activators of the voltage-gated Na+ channels. They may affect the AChR density in the brain by increasing neural (presynaptic) activity. Eriksson and Nordberg discuss this explicitly in their article as one putative mode of action (p. 461). (2) The effects of pyrethroids on AChR are also a matter of dispute. Burns et al. discuss opposing views on the findings on pyrethroid effects on AChR density (p. 240f). Finally (3) all available studies were performed on developing mice. Our text deals with insect cholinergics and although a discussion, whether pyrethroids may affect AChR density via a presynaptic increase of excitation may be interesting by itself, it does not add relevant information to the scope of our review.

The title of Section 4. “Insecticides on larval and adult development” seemingly covers many insecticides. However, the section mainly discusses the neonicotinoid, only a few other insecticides were included. In addition, it seems to overlap with the topic of section 5. The authors might please reconsider and modify the titles of each section or add more section titles is fine. 

We changed the title of section 4 according to the reviewer’s suggestion. In this section 4 we list the described effects of neonicotinoids on honeybee development whereas in section 5 we ask for the mechanisms underlying these effects. Since we intended to provide a conceptual review we incorporated in sections 5 and 6 also several integrative interpretations on how the effects of the cholinergic system regulates e. g., honeybee physiology, development and behavior as well as incorporating disturbing effects of insecticides that target ACh-dependent signaling into this frame.

Ln 237-238  Authors mentioned ACh may play a role in this plasticity. Please provide more supported refs or explanations. Only the HPG size change and nurse bee might feed hive mates with jelly along with ACh can’t be valid pieces of evidence or supports.

We have rewritten the section in Ln 242ff according to the reviewer’s suggestion.

Ln 251-3 “Besides these morphological abnormalities, neonicotinoid exposure also impairs hypopharyngeal gland function, because acetylcholine secretion into the brood food diminishes after clothianidin or thiacloprid feeding” Please provide refs.

We added the reference here (Wessler et al. 2016)

Ln 241-242 “Several studies indicate that gland development can be regulated by the cholinergic system.” Please provide refs.

We deleted this sentence (see response to #2). The references to the effects of neonicotinoids on hypopharyngeal gland development are cited in Ln 258 ff.

The division of labor has been correlated to several pheromones, including  JH, vitellogenin, and insulin. It will be good to take all of them into consideration and put them into your manuscript and model.

Ament, S. A., Wang, Y., & Robinson, G. E. (2010). Nutritional regulation of division of labor in honey bees: toward a systems biology perspective. Wiley Interdisciplinary Reviews: Systems Biology and Medicine, 2(5), 566-576.

 We rewrote the respective section and included the suggested review into the text. Thanks for the valuable hint (see response to #2 and #4).

There are other pesticides that can also induce the HPG alterations (review in Berenbaum et al 2019). Thus, they might affect the secretion of jelly and larvae development. Those pesticides and their effects would be good to mention in section 4.

Berenbaum, M. R., & Liao, L. H. (2019). Honey Bees and Environmental Stress: Toxicologic Pathology of a Superorganism. Toxicologic pathology, 0192623319877154.

 We added some sentences and references at the end of chapter 4 (ln. 273ff) to provide a more comprehensive overview.

The Fig.1 model was too simplified and could confuse readers. Does the model represent the adults or larvae system? Where is AChE or AChT? Could the authors add more details that you have mentioned in the text? “Gene expression” is not clear enough. Please indicate which genes might be involved. Do they down-regulate or up-regulate? How they associate with physiological and behavioral consequences if any?

The evidence, so far, indicated that the neuronal and non-neuronal AChRs lead to different responses. Could the authors separate the models here?  

Second, “+” and “-” is not clearly indicated in the figure caption. Please add them.

The figure summarizes and conceptualizes the described effects and ideas outlined in detail in the text. We extended the legend according to the suggestions but did not add additional figure elements since we are concerned that this would reduce the understandability of the figure. All effects of the cholinergic system (blockade of AChE, synthesis of ACh etc.) ultimately converge onto changes in AChR activation. In our opinion, including solely the AChR is sufficient to describe the effects and is in harmony with our focus on the receptors. However, we have only sparse information of which receptors are exactly involved in particular effects and where they are expressed, thus no separation into neuronal or non-neuronal system is possible today.

9. The References and their format will need further proofreading. There are some errors. Here only a part of them is listed. Please double-check all references and their format e.g., Ref. 2 and Ref. 73 were duplicated. Ref. 2. Volume and issue missed. Ref. 4 titles should use the sentence capital not capitalized all words.

All references have been corrected.